# Improved Detection of Herpesviruses from Diluted Vitreous Specimens Using Hydrogel Particles

**DOI:** 10.3390/diagnostics12123016

**Published:** 2022-12-01

**Authors:** Nicole L. Belanger, Robbie Barbero, Robert Barclay, Benjamin Lepene, Lucia Sobrin, Paulo J. M. Bispo

**Affiliations:** 1Department of Ophthalmology, Massachusetts Eye and Ear Infirmary, Boston, MA 02114, USA; 2Infectious Disease Institute, Harvard Medical School, Boston, MA 02115, USA; 3Ceres Nanosciences, Inc., Manassas, VA 20110, USA

**Keywords:** uveitis, herpesviruses, PCR, nanoparticles

## Abstract

Infectious uveitis is a sight-threatening infection commonly caused by herpesviruses. Vitreous humor is often collected for molecular confirmation of the causative agent during vitrectomy and mixed in large volumes of buffered saline, diluting the pathogen load. Here, we explore affinity-capture hydrogel particles (Nanotrap^®^) to concentrate low abundant herpesviruses from diluted vitreous. Simulated samples were prepared using porcine vitreous spiked with HSV-1, HSV-2, VZV and CMV at 10^5^ copies/mL. Pure undiluted samples were used to test capturing capability of three custom Nanotrap particles (red, white and blue) in a vitreous matrix. We found that all particles demonstrated affinity to the herpesviruses, with the Red Particles having both good capture capability and ease of handling for all herpesviruses. To mimic diluted vitrectomy specimens, simulated-infected vitreous were then serially diluted in 7 mL TE buffer. Diluted samples were subjected to an enrichment protocol using the Nanotrap Red particles. Sensitivity of pathogen detection by qPCR in diluted vitreous increased anywhere between 2.3 to 26.5 times compared to non-enriched specimens. This resulted in a 10-fold increase in the limit of detection for HSV-1, HSV-2 and VZV. These data demonstrated that Nanotrap particles can capture and concentrate HSV-1, HSV-2, VZV and CMV in a vitreous matrix.

## 1. Introduction

Uveitis is a serious and sight-threatening inflammation believed to be associated with 10% of legal blindness in the United States [1]. There are two major etiologies: infectious and non-infectious (immune-mediated), with infectious etiologies comprising around 20% of cases in the United States [2]. The spectrum of microbes causing infectious uveitis ranges depending on a variety of factors such as climate and regional distribution of pathogens, as well as the type of uveitis. The herpes viruses are common causative viruses in infectious posterior uveitis, these include cytomegalovirus (CMV), which causes retinitis in immunosuppressed patients, and herpes simplex types 1 and 2 (HSV-1 and HSV-2) and varicella zoster virus (VZV), which are commonly associated with acute retinal necrosis [3]. These viruses are also associated with anterior uveitis and are the most common etiologies in the western world [4]. CMV retinitis is the most common opportunistic ocular infection in patients with acquired immunodeficiency syndrome (AIDS) [5]. In patients who are immunocompetent, most cases of viral infectious uveitis are caused by HSV and VZV [6].

Laboratory diagnosis of infectious uveitis often relies on the use of polymerase chain reaction (PCR) testing of aqueous and vitreous humor with a significant percentage of false negatives among these tests in some cases due to a short-lived release of virus into the intraocular fluids or a viral load that is too low to be detected. The positive rates can range from 30% to 70% [7,8,9,10], with variations based on a variety of factors such as the stage of infection, type of infection, and population tested. In many cases, vitreous fluid is collected through vitrectomy and diluted in large volumes of buffered saline, which can substantially decrease the viral load resulting in poor sensitivity of pathogen detection.

The Nanotrap particles developed by Ceres Nanosciences are customizable hydrogel particles with diameters typically less than 1 micrometer. Hydrogel particles have many highly desirable characteristics for diagnostic use, such as: stability, uniformity, high surface area, easy suspendability in aqueous media, and versatility with regard to the ease of making physical-chemical modifications to the particles [11]. They can be engineered to have features such as: a charged or inert polymer shell, charged functional groups, and/or affinity dye functional groups [12].

Nanotrap particles have been tested against a variety of arthropod-borne viruses (arboviruses) such as Rift Valley fever virus (RVFV), *Venezuelan equine encephalitis* virus (VEEV), *Zika* virus (ZIKV), *Chikungunya* virus (CHIKV), and *Dengue* virus (DENV) [13,14]. It has been shown that Nanotrap particles capture and concentrate RVFV particles resulting in 100-fold greater enrichment of low viral titers that may show up as false negatives when processed without any form of enrichment [14]. Nanotrap particles have also enhanced the detection of ZIKV down to 10^2^ PFU/mL (plaque-forming units/mL) and CHIKV down to 10^1^ PFU/mL from urine samples [13]. Nanotrap particles have shown ability to capture human immunodeficiency virus type 1 (HIV-1) proteins, as well as the viral transactivator Tat, Nef, and gp41 from complex body fluids such as human serum, cerebrospinal fluid, and urine [15]. There has also been research showing that these particles are able to capture respiratory pathogens such as influenza A and B, respiratory syncytial virus (RSV) and coronavirus (CoV) [12]. Here, we explore the ability of Nanotrap particles to capture and enhance the viral load of dilute vitreous fluids infected with HSV-1, HSV-2, VZV and CMV, the most common causes of viral uveitis.

## 2. Materials and Methods

### 2.1. Control Viral Panels

Herpes viral verification panels for CMV, HSV-1, HSV-2, and VZV were obtained from Exact Diagnostics (SKU #CMVP100, HSV1P100, HSV2P100, VZVP100).

### 2.2. Sampling of Porcine Vitreous

Porcine eyes obtained from Sierra for Medical Science were placed into a beaker filled with 3% povidone iodine and soaked for 5 min. The eyes were then washed with 1× phosphate-buffered saline (PBS) until all the povidone iodine was flushed out. The eyes were then placed onto sterile square plates. Vitreous fluid was extracted using a 30-gauge needle attached to a 1 mL syringe. Vitreous was pooled together into a 50 mL conical tube. The solution was centrifuged 2 times for 5 min each at 2350× *g* (5000 rpm) to bring any remaining tissue to the bottom of the tube. After centrifuging, vitreous was transferred to a clean 50 mL conical tube and stored at −20 °C until later use.

### 2.3. Comparison of Three Different Nanotrap Particle Types

Three types of custom hydrogel particles were provided by Ceres Nanosciences (Manassas, VA, USA), each with different affinity dye functional groups. Nanotrap particle identifiers were created based on the visual color of the suspension for each particle type and are named as follows: Red, White, and Blue particles. Herpes viral verification panels for CMV, HSV-1, HSV-2, and VZV obtained from Exact Diagnostics were used to spike porcine vitreous at a concentration of 10^5^ cp/mL (copies per milliliter) or IU/mL (International Units per milliliter for CMV). Next, 100 µL of the newly created porcine solution was mixed with 100 µL of TE Buffer. The entire solution (200 µL) was added to each type of Nanotrap particle that had already been pelleted (in triplicate for each virus). After incubation, solutions were centrifuged to pellet the captured viral particles and the supernatant was discarded. Pellets were resuspended in 200 µL Buffer AL (QIAGEN, Hilden, Germany). Extraction of viral DNA was performed using the QIAamp^®^ DNA Mini Kit (QIAGEN) and eluted DNA was used for pathogen detection and quantification using a previously validated assay [16]. Each reaction contained 12.5 µL of Express SYBR™ GreenER™ (Thermo Fisher, Waltham, MA, USA), 0.5 µL of the appropriate forward and reverse primers, at 20 µM working stocks, 2.5 µL of template DNA, and 9 µL of nuclease-free water. The reaction was set up in at the following cycling conditions: 50 °C for 2 min, 95 °C for 5 min, and then 40 cycles of 95 °C for 5 s, 60 °C for 10 s (acquiring on green), followed by a high-resolution melt (HRM) that went from 75–95 °C by 0.1 °C/cycle. The gain was set to optimize each run. Results were analyzed using the Rotor-Gene^®^ Q Series Software (version 2.3.5) (QIAGEN). All PCR reactions were performed with no-template controls (NTCs).

### 2.4. Preparation of Simulated Diluted Vitreous Samples

Simulated diluted vitreous samples that mimic vitrectomy specimens were created by diluting (1:10) pure vitreous containing 10^5^ cp/mL (IU/mL for CMV) of viruses in TE buffer (Tris-EDTA, pH = 8.0) to create a 7 mL solution containing 10^4^ cp/mL. Serial dilutions were performed down to 10^1^ cp/mL. Prior to beginning the enrichment, 200 µL of each dilution (10^4^ cp/mL to 10^1^ cp/mL) was placed into a separate 1.5 mL tube in triplicate and set aside to be extracted as the non-enriched samples as shown in Figure 1.

### 2.5. Viral Capture and Concentration with Nanotrap Red Particles

To begin the enrichment process, 180 µL of the Nanotrap Red particles were pipetted into 2 mL tubes (3 for each dilution) and centrifuged at 15,000× *g* for 10 min to pellet the particles. Once centrifugation was completed, the supernatant was removed carefully as to not disturb the pellet. In triplicate, 1.8 mL of the diluted vitreous samples was added to each tube. The solution was pipetted up and down to resuspend the pellet, careful not to directly poke the pellet. The tubes were then incubated at room temperature for 10 min, making sure to invert the tubes to mix the solution every 5 min. When incubation was complete, the solutions were centrifuged at 15,000× *g* for 10 min. After centrifugation, the supernatant was removed. The pellet was then resuspended in 200 µL of nuclease-free water. Viral DNA was then extracted using the Zymo Quick-DNA Microprep Plus Kit (Zymo Research, Irvine, CA, USA). For both the enriched and non-enriched samples, 800 µL of Genomic Lysis Buffer (with β-mercaptoethanol) was added to each of the tubes, vortexed, and set to incubate at room temperature for 5 min. After incubation was complete, the solution was added to the Zymo-Spin IC-XM Columns in collection tubes and centrifuged at 10,000× *g* for 1 min, discarding the supernatant. This was done by adding the solution in 2 increments of 500 µL as to not overflow the columns. Next, 200 µL of the DNA Pre-Wash Buffer was added to the columns and centrifuged at 10,000× *g* for 1 min, discarding the supernatant. Lastly, 500 µL of the g-DNA Wash Buffer was added to each of the columns and centrifuged at 10,000× *g* for 1 min, discarding the supernatant. The columns were then added to clean, 1.5 mL microcentrifuge tubes. Then, 20 µL of DNA Elution Buffer was added directly to the columns and set to incubate at room temperature for 10 min. DNA was eluted by centrifuging at 10,000× *g* for 1 min. PCR was performed in duplicate for each of the triplicate samples at the 4 dilutions (10^4^ cp/mL to 10^1^ cp/mL) using real-time PCR assays (described above). The process of enrichment, entrapment, and DNA extraction is summarized in Figure 2 and Figure 3. Non-enriched aliquots (200 µL) were extracted in the same manner. PCR for enriched and non-enriched samples was performed in duplicate for each of the triplicate samples at the 4 dilutions (10^4^ to 10^1^) as described above. All PCR reactions were performed with no-template controls (NTCs).

### 2.6. Construction of Reference Curves

Pure solutions of viral particles from Exact Diagnostics were created for each virus from the concentrations of 10^5^ to 10^1^ (cp/mL or IU/mL for CMV). To do this, the stock solution of 10^6^ was diluted 1:10 in TE Buffer so that a 200 µL solution for each desired concentration was made. The extraction was performed in the same method as described above using the Zymo Quick-DNA Microprep Plus (Zymo Research). After DNA was eluted, the same PCR was performed as mentioned in the previous section. The results of this PCR were used to import a reference curve into the PCR runs testing the enriched vs. non-enriched samples at each concentration. This allowed for the software to estimate the concentration of each sample run based on the set standard concentration of the pure solutions used in the run to create the reference curve. The VZV reference curve was constructed using a VZV plasmid which was cloned using the Zero Blunt™ TOPO™ PCR Cloning Kit for Sequencing (Thermo Fisher).

### 2.7. Statistical Analyses

GraphPad Prism 9 version 9.3.1 (471) was used for the construction of graphs and statistical analyses. For the three different particle types (Red, White, and Blue), an ordinary one-way ANOVA was run to compare the performance of each particle type to one another for the four viruses tested. To understand the differences between both enriched (with the Red particles) and non-enriched samples at the four concentrations tested for each virus tested, a two-way ANOVA was run.

## 3. Results

### 3.1. Performance of Each Particle Type

To evaluate the ability of three different custom Nanotrap particles to capture uveitis-causing herpesviruses in a vitreous matrix, simulated samples at 10^5^ cp/mL were incubated with each of the tested Nanotrap particles. All Nanotrap particles were capable of capturing HSV-1, HSV-2, VZV and CMV spiked in a vitreous matrix. An ordinary one-way ANOVA was used to compare the three different particle types. For CMV, HSV-2, and VZV, there was a statistically significant (*p* < 0.05) difference between the particle types (*p* = 0.0024, *p* = 0.0073, and *p* < 0.0001, respectively). For HSV-1, the difference between the particle types did not appear to be statistically significant (*p* = 0.0639). Because the Red particles displayed good affinity to all herpesviruses tested and was easier to handle (compared to the White particles, which formed hard pellets that were difficult to resuspend), they were chosen for the enrichment experiments that follow. Mean and standard deviation data for each particle type and virus is shown in Table 1. Results of each particle enrichment experiment are shown in Figure 4.

### 3.2. Comparison of Enriched vs. Non-Enriched Samples

We found that adding an enrichment step with this novel hydrogel particle can concentrate all uveitis-causing herpesviruses tested and enhance the sensitivity of detection by PCR.

At a viral load of 10^4^ cp/mL (or IU/mL for CMV), the Nanotrap Red provided a 3- and 15-fold increase in detection of HSV-1 and HSV-2, respectively. At lower concentrations, the fold increase ranged from 2.5–2.7 for HSV-1 and 13.6–26.5 for HSV-2 (Figure 5, Table 2). This resulted in limit of detections (LoD) for enriched samples that were 10-fold higher when compared to non-enriched specimens (Table 2). Concentration of viral particles using the Nanotrap Red particles increased the LoD for HSV-1 and HSV-2 from 10^2^ cp/mL to 10^1^ cp/mL. Similarly, a 3.2 to 5.7-fold increase in detection of VZV was obtained for enriched samples, resulting in a 10-fold improvement in the LoD (from 10^3^ cp/mL to 10^2^ cp/mL). For CMV, enrichment with the Nanotrap Red resulted in a 2.3- to 11.1-fold increase in detection (Figure 5, Table 2). Despite of this improvement in signal detection for CMV, the LoD of detection remained the same before and after enrichment for the log10 dilutions included in our study. A two-way ANOVA analysis was performed to compare the enriched and non-enriched samples at each concentration of viral load. The difference for each herpesvirus was found to be statistically significant with a *p*-value of *p* < 0.0001.

## 4. Discussion

Because many samples from uveitis patients that are processed for viral detection and diagnosis confirmation are collected thorough vitrectomy and diluted with large volumes of buffered saline, the number of viral particles available for detection in a small aliquot of dilute vitreous can be substantially decreased. Concentrating these viral particles present in milliliters of dilute vitreous into a minute amount (microliters) that can be processed using the current DNA preparation methods could significantly increase the sensitivity of detection by PCR and improve diagnostic yields. To test this hypothesis, we quantified the viral loads of simulated-infected dilute vitreous samples before and after an enrichment process using the Nanotrap Red participles. The Nanotrap particles have shown to have good binding affinity to HSV-1, HSV-2, VZV and CMV in a vitreous matrix. The enrichment with Nanotrap particles increased viral detections in dilute samples at the highest concentration tested (10^4^). Enrichment also enabled detection by PCR for samples at the lower concentrations of 10^1^ and 10^2^. For HSV-1, HSV-2, and VZV, samples enriched with Nanotrap particles had a 10-fold increase in the LoD compared to samples that were not enriched. Enrichment of CMV with Nanotrap particles increased the signal for detectable dilutions, while the LoD remained unaltered for the small range of log10 dilutions tested. Since an improvement in detection was still detected for CMV after enrichment with Nanotrap Red, it is likely that the LoD would be impacted if intermediate dilutions between 10^1^ and 10^2^ were tested.

PCR testing is commonly used for pathogen detection in patients presenting with infectious uveitis [17,18,19,20,21,22]. Real-time PCR is a method often explored for development of sensitive and rapid assays to detect the presence of herpesviruses from aqueous and vitreous samples [18,19] and in some cases from whole blood and cerebrospinal fluid (CSF) or serum [17,23]. Aqueous or vitreous are typically the desired specimen to test for ocular disease as PCR testing done on serum can identify concomitant infections present within the individual and not accurately diagnose the infection associated with uveitis [23]. Oftentimes, diagnostic aqueous or vitreous taps for specimen collection only yield about 50–200 µL of ocular fluids. Aqueous and vitreous taps are the most frequently used diagnostic procedures for obtaining ocular fluids because they can be easily done in the outpatient clinic setting, and between the two aqueous tap is most frequently employed because it is more likely to give a yield, technically easier and more comfortable for the patient. When vitreous is acquired through vitrectomy, the volume is larger than a vitreous tap. Part of the specimen is usually obtained undiluted (typically about 1 mL), but a larger, diluted vitreous sample can also be obtained. This diluted sample is often significantly diluted in buffered saline used as the infusion fluid for the vitrectomy, and this substantially dilutes the pathogen responsible for infection in the sample. Our validated PCR assay [16] was efficient at detecting concentrations of 10^2^ cp/mL for HSV-1, HSV-2 and CMV viral particles, and 10^3^ cp/mL for VZV. With the addition of Nanotrap particles, we were able to concentrate low viral titers of sample (10^2^ and 10^1^) in simulated vitrectomy specimens, so they were still detectable by PCR at copy numbers as low as 10^2^ cp/mL for CMV and VZV, and even at 10^1^ cp/mL for HSV-1 and HSV-2, allowing us to detect herpesviruses below the established LoD of our real time PCR assay for samples that are not concentrated and enriched. This aligns with what other studies have done regarding the use of Nanotrap particles for enrichment of a variety of viruses in clinical specimens [11,12,13,14,15], which demonstrated the usefulness of these particles to entrap and concentrate viruses that are associated with significant disease burden in humans such as arboviruses [13,14] and respiratory viruses [12]. Here, we have shown that Nanotrap particles are also capable of binding to herpesviruses that are widely disseminated in the human population and are common culprits of sight-threatening intraocular infections and demonstrated that these particles can be used to concentrate the viral load and improve PCR detection in simulated vitrectomy specimens that are substantially diluted large volumes of buffered saline.

Routinely, diagnosis of uveitis is made on the basis of the clinical presentation. However, determining the etiology can be complicated by overlapping findings among cases caused by different pathogens and also with the features found for noninfectious uveitis cases [24]. This can delay the initiation of appropriate therapy and result in poor outcomes [6]. Molecular detection of herpesviruses and other pathogens by PCR testing of intraocular fluids is a common approach for etiological diagnosis of infectious uveitis. Although these tests have in general good analytical sensitivity, there are false negative results for a considerable number of samples [17].

False negatives can occur even if samples are collected under optimal conditions—with an undiluted, but the reduced clinical sensitivity of PCR for detection of a causative agent can be, in part and in some cases, related to the use of highly diluted vitreous specimens collected through vitrectomy, which contain very low viral titers [7]. Diluted vitreous is sometimes sent for PCR instead of undilute vitreous because the cause is uncertain at the time of vitrectomy between a masquerading malignancy and an infectious cause, and the undilute specimen is prioritized for tests to detect malignancy. Capture and concentration of viral particles could potentially improve the sensitivity of PCR and result in increased diagnostic yields. The use of Nanotrap particles for pathogen enrichment has been shown to substantially improve the detection of respiratory viruses [12] and arboviruses [13,14] in complex clinical matrices, and its use in the molecular diagnostics of infectious uveitis could potentially help improve detection rates and clinical sensitivity of PCR tests in specimens with low viral titer. Another clinical scenario where Nanotrap particles could be useful, which we did not explore in this paper, is in partially treated patients. Often patients are referred already having started an empiric course of antiviral therapy, but an inadequate response or other clinical features bring the diagnosis into question and a diagnostic ocular fluid is then procured. Aqueous and vitreous fluid often have a lower yield for PCR in these partially treated patients and Nanotrap particles could improve the yield here as well. The role of Nanoparticles in partially treated ocular fluid samples, as well as in aqueous samples, a more common ocular fluid used for diagnosis of viral posterior uveitis despite typically having a lower viral load than vitreous, are areas for future studies with this technology.

## 5. Conclusions

Nanotrap Red particles were successful at trapping and enhancing detection of viral particles in a vitreous matrix and simulated diluted vitrectomy specimens. Our data supports the use of the Nanotrap Red particles for pre-processing of diluted vitreous specimens for improved detection of CMV, HSV, and VZV by PCR. Additionally, the use of Nanotrap particles will only slightly increase the sample processing time by about 30–60 min. The average sample costs about $10 USD to extract and perform PCR. With the addition of Nanotrap particles, the cost would increase to approximately $15 USD per sample. The advantages of this sample enrichment process are expected to translate into improved diagnostic yields for specimens collected from patients presenting with viral uveitis and processed by PCR, ultimately helping to improve care and clinical management.

## Figures and Tables

**Figure 1 diagnostics-12-03016-f001:**
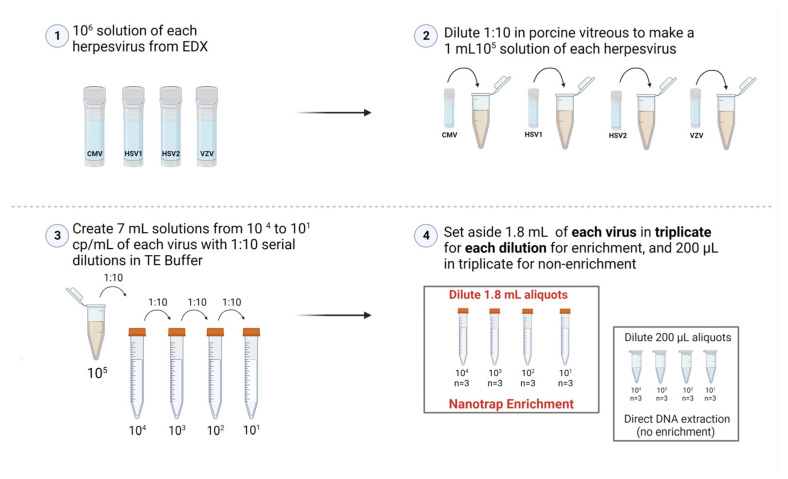
Creating simulated vitreous samples. The serial dilution process to prepare samples to be either enriched with Nanotrap Red particles or not enriched (Created with BioRender.com).

**Figure 2 diagnostics-12-03016-f002:**
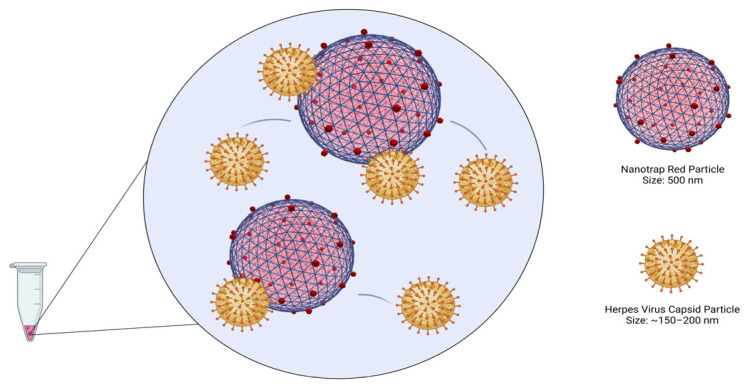
Binding of herpesviruses to the Nanotrap Red particles (Created with BioRender.com).

**Figure 3 diagnostics-12-03016-f003:**
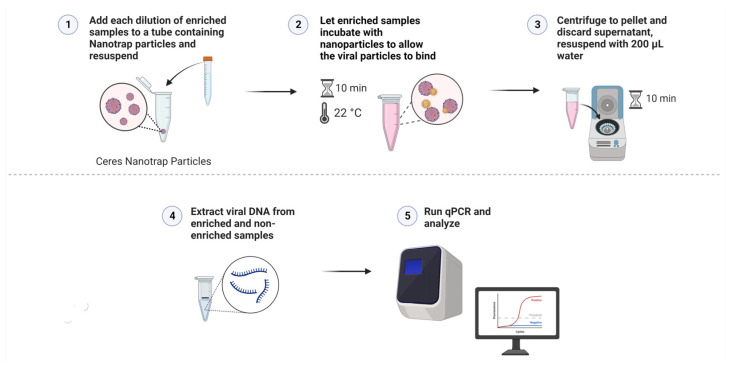
A visual protocol demonstrating the use of Nanotrap Red particles to enrich the samples and the processes of DNA extraction and PCR that followed (Created with BioRender.com).

**Figure 4 diagnostics-12-03016-f004:**
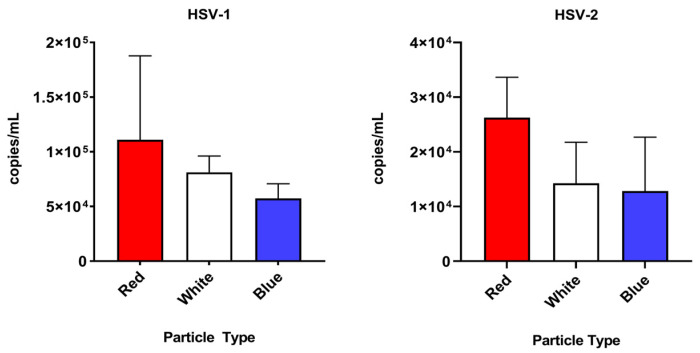
A comparison of the three particle types tested (Nanotrap Red, Nanotrap White, and Nanotrap Blue) against the four herpesviruses tested in this study.

**Figure 5 diagnostics-12-03016-f005:**
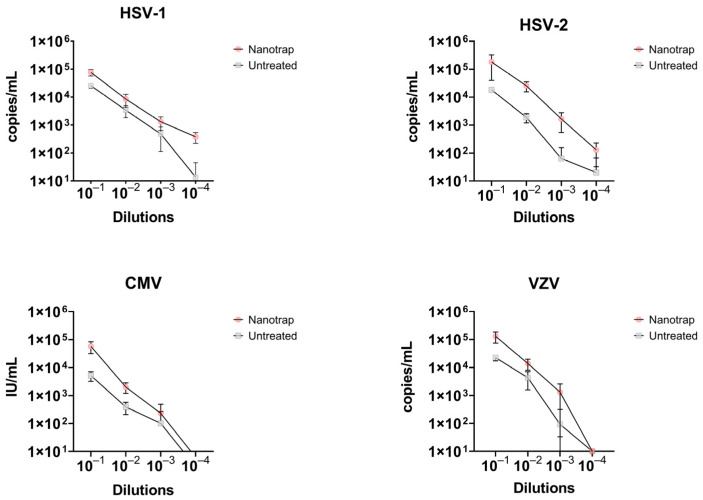
A comparison of samples enriched with Nanotrap Red particles versus those untreated in terms of copy number per mL.

**Table 1 diagnostics-12-03016-t001:** Mean and standard deviation values (in cp/mL) for each of the custom Nanotrap particle types tested: Nanotrap Red, Nanotrap White, and Nanotrap Blue.

	Nanotrap Red		Nanotrap White		Nanotrap Blue	
	Mean *	SD	Mean	SD	Mean	SD
CMV	34,157.8	15,194.1	33,502.2	144,512	13,454.8	6271.9
HSV-1	110,954.9	76,777.7	81,236.4	14,932	57,353.6	13,377.8
HSV-2	26,255.2	7395.3	14,253.5	7496.8	12,822.3	9859.6
VZV	171,683.8	79,047.8	178,146.5	41,784.4	31,991.6	25,461.7

* All mean values have concentration given in cp/mL except for CMV which is in IU/mL.

**Table 2 diagnostics-12-03016-t002:** Mean and standard deviations (in cp/mL), as well as mean Ct value, for both the enriched and non-enriched samples of each pathogen at the 4 concentrations tested (10^4^ to 10^1^).

		Enriched			Non-Enriched			Fold Increase
		Mean *	SD	Mean C_t_	Mean	SD	Mean C_t_	
HSV-1	10^4^	76,252.5	19,527.4	23.96	25,351.6	4671.6	25.58	3.0
	10^3^	8484.1	4075.9	27.31	3397.3	1559.0	28.68	2.5
	10^2^	1303.1	657.7	30.16	480.5	367.3	32.25	2.7
	10^1^	380.2	160.3	31.95	0 ^§^	-	>40	-
HSV-2	10^4^	276,249.6	22,581.5	22.67	18,241.9	1953.7	26.60	15.1
	10^3^	25,437.6	10,008.6	26.20	1863.7	679.14	29.98	13.6
	10^2^	1638.9	1105.1	30.62	61.9	89.6	37.46	26.5
	10^1^	126.9	95.5	34.20	0	-	>40	-
CMV	10^4^	57,465.4	25,857.7	23.71	5162.7	1948.1	27.42	11.1
	10^3^	2022.9	825.4	28.86	390.5	182.3	31.41	5.2
	10^2^	230.9	258.13	33.76	101	170.6	37.12	2.3
	10^1^	0	-	>40	0	-	>40	
VZV	10^4^	130,149.9	56,026.1	29.06	22,669.5	5395.1	31.48	5.7
	10^3^	13,950.6	5956.7	32.28	4313.8	2735.0	34.34	3.2
	10^2^	1303.9	1270.9	36.91	0	-	>40	-
	10^1^	0	-	>40	0	-	>40	-

* All mean values have concentration given in cp/mL except for CMV which is in IU/mL. ^§^ A mean value of 0 indicates that only 1 or 2 of the 6 replicates amplified.

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
