# Peer review of "Improved Detection of Herpesviruses from Diluted Vitreous Specimens Using Hydrogel Particles"

_diagnostics, 2022, doi:10.3390/diagnostics12123016_

Round 1

Reviewer 1 Report

Comments to the Author:

The overall design of this study is reasonable and the paper demonstrated that Nanotrap particles can capture and concentrate herpesvirus in simulated diluted vitrectomy specimens. However, there are several points which could be improved.

1. Line 77-85 mentioned that sampling of porcine vitreous, in the experiment, how many porcine eyes in total were used? And how many porcine eyes were used to create each viral vitreous?

2. The authors used three types of custom hydrogel particles, each with different affinity dye functional groups. Are the three custom hydrogel particles merely differences in color? Did the different affinity dye functional groups have different structures?

3. Line 121-150 only referred to the use of Nanotrap Red particles for viral capture and concentration. Why did the article not mention White and Blue particles? And 189-190 mentioned that compared with White particles, Red particles are easier to handle. I suggest adding the similarities and differences of the three particles experimental steps in the methods.

4. It is inappropriate to compare the three particle types of each herpesvirus by unpaired t-tests.

5. The Results section should be based on objective data, it is suggested to modify the expression of line 202-212.

6. The authors stated that the Nanotrap Red particles increased the LoD from 102 cp/mL to 101 cp/mL, this expression is inaccurate. There is the same error in line 220.

Reviewer 2 Report

The manuscript by Belanger et al. provides interesting and well-organized study. Molecular enrichment techniques in low-abundant clinical samples can vastly improve diagnostics and decrease the number of false-negative results. The authors applied proprietary Nanoparticles from Ceres Nanosciences, Inc., to simulate capturing and enhancement of the viral load in dilute vitreous fluids infected with HSV-1, HSV-2, VZV and CMV, the most common agents causing viral  uveitis. Aqueous and vitreous fluid often have a lower yield for PCR and used Nanoparticles could improve the yield here as well.
I have several questions regarding this manuscript.

Abstract. Line 14. 105  copies/mL or 1E+05 (105) copies/mL ?

1. Introduction.

Line 63. 102 PFU/mL means 1E+02 (102) PFU/mL ? Same on Line 64. 101 PFU/mL ?

2. Materials and Methods.

Please, indicate the subsections of this section, eg 2.1, 2.2. etc.

Figure 2. I'm not sure that this figure is informative enough, because it does not reflect the capture mechanism of the virus by the nanoparticle. Do the authors have data on the maximum "capacity" of a single Nanoparticle in relation to each of the HSV-1, HSV-2, CMV, VZV? It can be interesting, since authors obtained impressive difference between HSV-2 and HSV-1 quantitative detection during the evaluation of enrichment at different dilutions.

Since authors use different statistical tests (unpaired t-tests, ANOVA) it would be useful to create corresponding subsection in the ‘Materials and methods’, for example, ‘Statistical data processing’.

3. Results.

Section 3.1.  What initial concentrations of viruses were used in this experiment? How was the effectiveness of virus capture by nanoparticles assessed? Nanoparticles were added to the initial concentration of viruses, carried out through the procedure of DNA isolation and quantitative PCR, as indicated in Methods ? If so, it would be interesting to see the control column (no nanoparticles added) on Figure 4 for each virus.

Section 3.2. Table 2. Authors obtained impressive difference in fold increase between HSV-2 (104-102) and HSV-1 (104-102). The HSV2 and HSV1 are really similar. Can authors explain these data ? Maybe some words in Discussion about this result ?

4. Discussion. Line 243 – (104) , should be 1E+04 (104) copies/mL ? Same for line 244 – «101 and 102» ?

This section is well writen. Authors provide the fields of application of Nanotrap particles. Any limitations of this study ? How can the use of nanoparticles be integrated into automated systems for processing clinical samples and isolating DNA from pathogens ?

Please, correct 'HSV1' and 'HSV2' to 'HSV-1' and 'HSV-2' through the text (see HSV1 and HSV2 on Figures, and in the Table 2).
